# Asymmetric and adaptive reward coding via normalized reinforcement learning

**Kenway Louie**[1,2] *

**1** Center for Neural Science, New York University, New York, United States of America, **2** Neuroscience Institute, New York University Grossman School of Medicine, New York, United States of America

* klouie@cns.nyu.edu

## Abstract

Learning is widely modeled in psychology, neuroscience, and computer science by prediction error-guided reinforcement learning (RL) algorithms. While standard RL assumes linear reward functions, reward-related neural activity is a saturating, nonlinear function of reward; however, the computational and behavioral implications of nonlinear RL are unknown. Here, we show that nonlinear RL incorporating the canonical divisive normalization computation introduces an intrinsic and tunable asymmetry in prediction error coding. At the behavioral level, this asymmetry explains empirical variability in risk preferences typically attributed to asymmetric learning rates. At the neural level, diversity in asymmetries provides a computational mechanism for recently proposed theories of distributional RL, allowing the brain to learn the full probability distribution of future rewards. This behavioral and computational flexibility argues for an incorporation of biologically valid value functions in computational models of learning and decision-making.

## Author summary

Reinforcement learning models are widely used to characterize reward-driven learning in biological and computational agents. Standard reinforcement learning models use linear value functions, despite strong empirical evidence that biological value representations are nonlinear functions of external rewards. Here, we examine the properties of a biologically-based nonlinear reinforcement learning algorithm employing the canonical divisive normalization function, a neural computation commonly found in sensory, cognitive, and reward coding. We show that this normalized reinforcement learning algorithm implements a simple but powerful control of how reward learning reflects relative gains and losses. This property explains diverse behavioral and neural phenomena, and suggests the importance of using biologically valid value functions in computational models of learning and decision-making.

## Introduction

Reinforcement learning (RL) provides a theoretical framework for how an agent learns about its environment and adopts actions to maximize its cumulative long-term reward [1]. In

**Data Availability Statement:** There are no primary data in the paper; simulation data and MATLAB code used for simulation and analysis of the NRL model are available at https://doi.org/10.17605/OSF.IO/E6T5Z.

**Funding:** The author received no specific funding for this work.

**Competing interests:** The authors have declared that no competing interests exist.

standard RL models, the values of stimuli or actions are learned via a reward prediction error (RPE), defined as the difference between actual and expected outcomes. Using RPE signals over repeated samples from the environment, learners progressively update their estimates to obtain a measure of the average value (e.g. of an action, a state, or a state-action pair). For example, in its simplest form an RL model updates its value estimate $V_{t+1}$ as:

$$V_{t+1} = V_t + \eta(R_t - V_t) \tag{1}$$

where the incremental update signal comprises a learning rate $\eta$ and an RPE term. The canonical RPE term is the difference between actual reward $R_t$ and expected reward $V_t$, which produces value signals that are a linear function of reward. RL algorithms accurately capture various types of learning in animal and human subjects [2,3], and have been employed in increasingly complex ways to produce powerful artificial intelligence agents [4–6]. Furthermore, the activity of midbrain dopamine neurons closely matches theoretical RPE signals [7–9], suggesting that RL serves as a computational model of neurobiological learning.

Despite their diversity and ubiquity, almost all RL approaches utilize a linear reward function at odds with evidence for nonlinear reward coding in the brain. A nonlinear relationship between internal subjective value and external objective reward is a longstanding assumption in psychology and economics. For example, risk preferences in choice under uncertainty are consistent with choosers employing a utility function that is a nonlinear function of reward [10,11]. This behavioral nonlinearity is reflected in underlying neural responses, with activity in reward-related brain areas correlating with subjective values rather than objective reward amounts [12–14]. Notably, in contrast to standard RL assumptions, the activity of dopamine neurons also exhibits a nonlinear response consistent with a subtraction between actual and expected reward terms, both of which are sublinear functions of reward amount [15,16]. Together, these results suggest that neurobiological RL mechanisms operate on nonlinear reward representations, but the behavioral and computational implications of nonlinear RL are not well understood.

Here, we develop and characterize an RL algorithm that learns a nonlinear reward function implemented via the divisive normalization computation [17]. Originally proposed to describe nonlinear responses in early visual cortex, normalization has been observed in multiple brain regions, sensory modalities, and species, suggesting that normalization is a canonical neural computation [18]. In addition to sensory processing, normalization explains neural responses in higher-order cognitive processes including attention, multisensory integration, and decision-making [19–22]. In particular, its role in reward coding makes it an attractive candidate mechanism for nonlinear reward representations in reinforcement learning.

## Results

### Normalized reinforcement learning model

In contrast to standard RL, we propose a normalized RL algorithm (NRL) that learns a value function nonlinearly related to objective reward. Specifically, NRL (Fig 1A) assumes that objective rewards $R$ are represented by a normalized subjective value function $U$:

$$U(R) = \frac{R^n}{\sigma^n + R^n} \tag{2}$$

and that learning employs the corresponding normalized prediction error term:

$$V_{t+1} = V_t + \eta(U(R_t) - V_t) = V_t + \eta\left(\frac{R_t^n}{\sigma^n + R_t^n} - V_t\right) \tag{3}$$

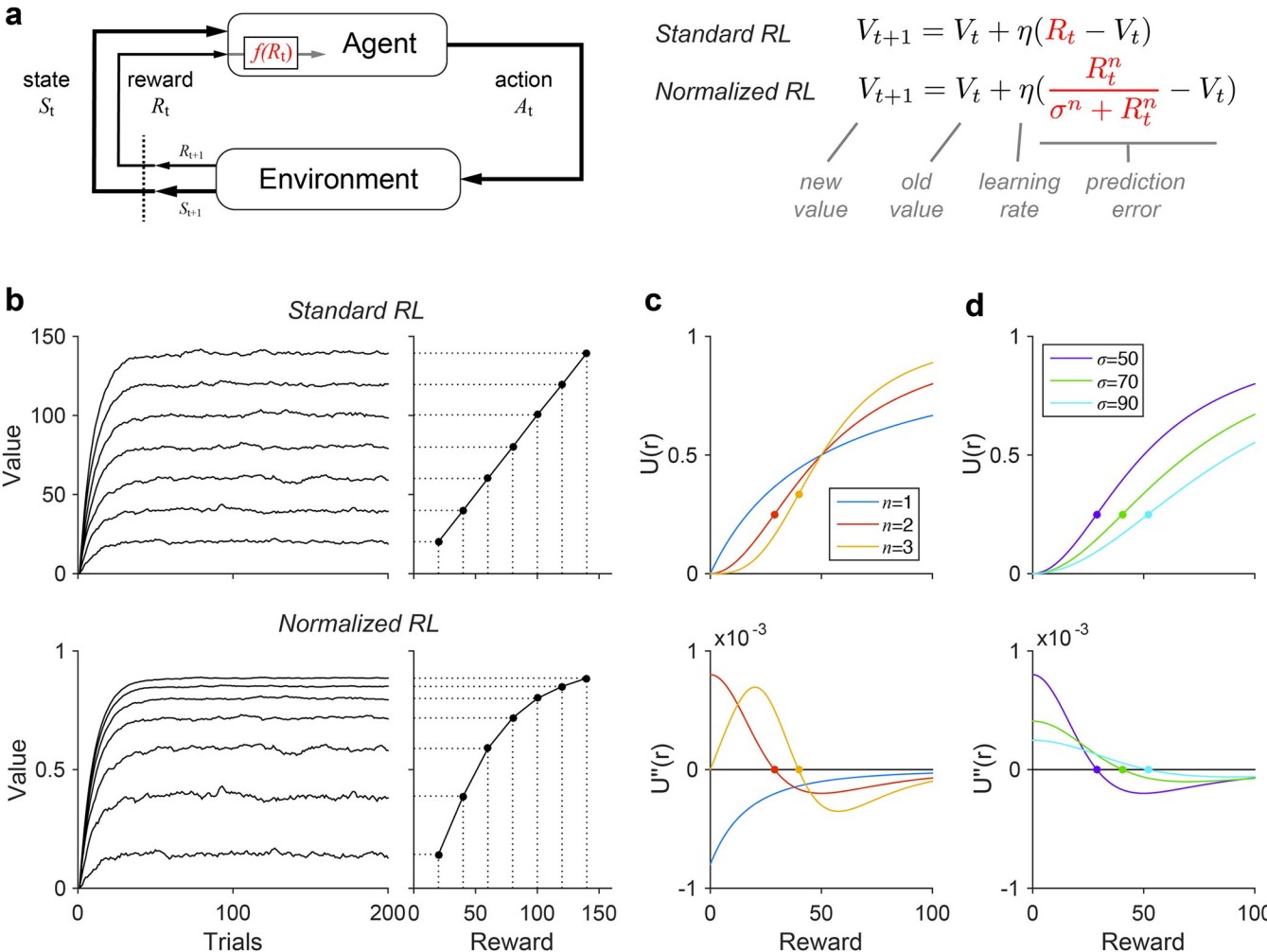

**Fig 1. Normalized reinforcement learning model. (a)** Comparison of standard reinforcement learning (RL) and normalized reinforcement learning (NRL) models. RL and NRL differ in how external rewards are transformed by the reward coding function $f(R_t)$ prior to learning internal value estimates. Standard RL uses a linear reward function, while NRL uses a divisively normalized representation. **(b)** Learned value functions under RL and NRL. Left, dynamic value estimates during learning. Right, steady state value estimates. Simulations were performed for each of seven rewards, with additive zero-mean Gaussian noise (learning rate $\eta = 0.1$). In contrast to RL, NRL algorithms learn values that are a nonlinear function of external rewards (example NRL simulation parameters: $\sigma = 50$, $n = 2$). **(c)** Convexity and concavity in NRL value functions. Top, NRL value functions with different exponents (fixed $\sigma = 50$ A.U.). Bottom, second derivative of value functions. Dots show inflection points between convex and concave value regimes. **(d)** Parametric control of NRL value curvature. NRL value function (top) and second derivative (bottom) for different $\sigma$ values (fixed $n = 2$).

where the exponent and semisaturation parameters $n$ and $\sigma$ govern the precise form of the nonlinear transformation (see below). Normalization produces a value function that is a saturating function of reward: at small reward magnitudes ($R << \sigma$) value grows with reward, while at large reward magnitudes ($R >> \sigma$) value approaches an asymptote. Consistent with this intuition, in simulation NRL algorithms learn values that are a saturating nonlinear function of reward (Fig 1B).

Unlike standard nonlinear utility functions, normalized value functions can exhibit an inflection point in reward responses that introduces an intrinsic, magnitude-dependent asymmetry in RPEs. The direction and extent of this asymmetry depends on the specific parameterization of the normalized value function. The exponent $n$ in normalization models governs amplification of inputs, and is generally considered a fixed property of the model [17,18]. For

$n \leq 1$, normalized value functions are always concave; however, for $n > 1$, normalized value functions are convex at lower rewards and concave at higher rewards (Fig 1C); convex (concave) regimes are evident as positive (negative) regions of the normalized value second derivative. In the convex regime, increases in $R$ generate larger changes in $U(R)$ than equivalent decreases in $R$; this predicts asymmetric RPE responses biased towards outcomes better than expected. In contrast, in the concave regime, decreases in $R$ generate larger $U(R)$ changes and RPEs are biased towards outcomes worse than expected. Notably, both theoretical and empirical considerations support normalization exponents consistent with inflection points: input squaring ($n = 2$) was used to model threshold linear responses in spiking activity in the original normalization equation [17], and fits of $n$ to neural data typically yield values between 1.0 and 3.5 (average value of 2) [18,23]. Thus, asymmetric RPE responses are likely to arise if reward learning relies on normalized value coding.

Critically, the type of NRL RPE asymmetry is parametrically tunable and depends on the relationship between rewards and the semisaturation term $\sigma$. Value function convexity and resulting positive-biased RPE asymmetry arise for rewards less than $\sigma$, while value concavity and negative-biased RPE asymmetry arise for rewards greater than $\sigma$. When $\sigma$ is varied, the reward amounts generating value concavity and convexity shift accordingly (Fig 1D; see S1 Appendix). As a result, asymmetric prediction errors around a given reward magnitude can be either negatively or positively biased, depending on the magnitude of the semisaturation term (Fig 2). Thus, the degree and direction of RPE asymmetry is parametrically controlled in the NRL algorithm; we next examine how variability in this asymmetry can generate variability in risk preferences (across individuals) and in reward learning (across information processing channels).

## Variability in risk preference

RPE asymmetry predicts that reinforcement learners will differentially weight outcomes that are better or worse than expected, consistent with behavior in multiple empirical studies [24–27]. However, previous studies–assuming linear reward coding—attributed this asymmetry to different learning rates for positive versus negative RPEs. For example, variable risk preferences in choice under uncertainty can be captured by standard RL models with valence-dependent learning rates [28]:

$$V_{t+1} = \begin{cases} V_t + \eta^+(R_t - V_t) \text{ if } (R_t - V_t) > 0 \\ V_t + \eta^-(R_t - V_t) \text{ if } (R_t - V_t) < 0 \end{cases} \tag{4}$$

For choices between a certain option and an uncertain option with the same mean outcome, these different learning rates implement a risk-sensitive RL process. If $\eta^- > \eta^+$, the model will learn an uncertain option value lower than its mean nominal outcome and exhibit risk-aversion; in contrast, if $\eta^- < \eta^+$, the model will overestimate uncertain options and exhibit risk-seeking behavior. RL models with differential learning rates have been increasingly influential and examined in the context of their adaptive properties [29,30], role in cognitive biases [27,31], and potential neurobiological substrates [24,25,32].

Here, we show that the NRL model can generate variable risk preferences without requiring different learning rates. We simulated NRL behavior in a dynamic learning and choice task, in which the value of certain and risky options had to be learned via outcomes [26]. Given the relationship between $\sigma$ and RPE asymmetry, the NRL model can generate either risk averse or risk seeking behavior (Fig 3A, blue lines; example risk averse $\sigma = 20$, example risk seeking $\sigma = 60$). Across a population of simulated subjects ($N = 50$), NRL agents generated a range of behavioral risk aversion levels that are strongly correlated with $\sigma$ ($r = -0.889$; $p = 7.08 \times 10^{-18}$).

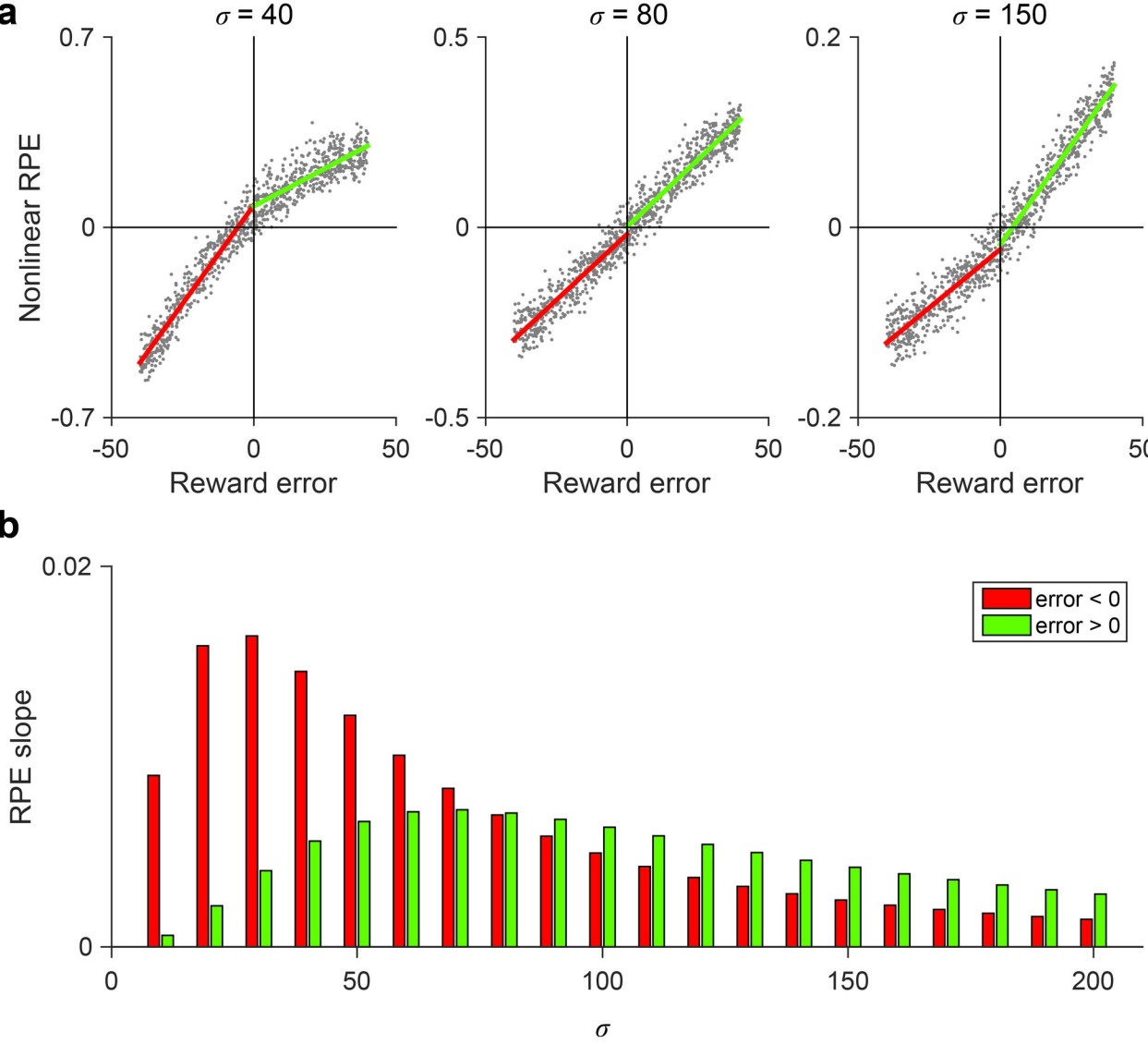

**Fig 2. Parametric control of prediction error asymmetry. (a)** Examples of variable reward prediction error (RPE) asymmetry. Each panel shows NRL responses for reward inputs ($R = 50$ A.U.) with uniformly distributed noise. Lines show piecewise linear regression fits for negative (red) and positive (green) reward errors. **(b)** The NRL semisaturation term governs the degree and direction of RPE asymmetry. NRL RPE asymmetry is biased towards negative RPEs at low $\sigma$ and postive RPEs at high $\sigma$.

However, if a linear reward function is wrongly assumed, RL models with differential learning rates (Eq 4) can accurately fit NRL-generated behavior (Fig 3A, black lines); furthermore, these data will appear to support differential learning rates linked to risk preference (Fig 3A, right). Intuitively, these apparent learning rates arise because fitting with the standard RL model (Eq 4) and two valence-dependent learning rates approximates bipartite linear regression on the nonlinear value function. In other words, the different curvature of the NRL value function for negative and positive RPEs (as seen in examples in Fig 2A) will be captured (incorrectly) as differential linear modulation by negative and positive learning rates. As a result, NRL-generated data will demonstrate a consistent relationship between apparent learning rate asymmetry and behavioral risk aversion (Fig 3B). However, this relationship is driven by the strong relationship between risk preference and the generating NRL model $\sigma$ (Fig 3C).

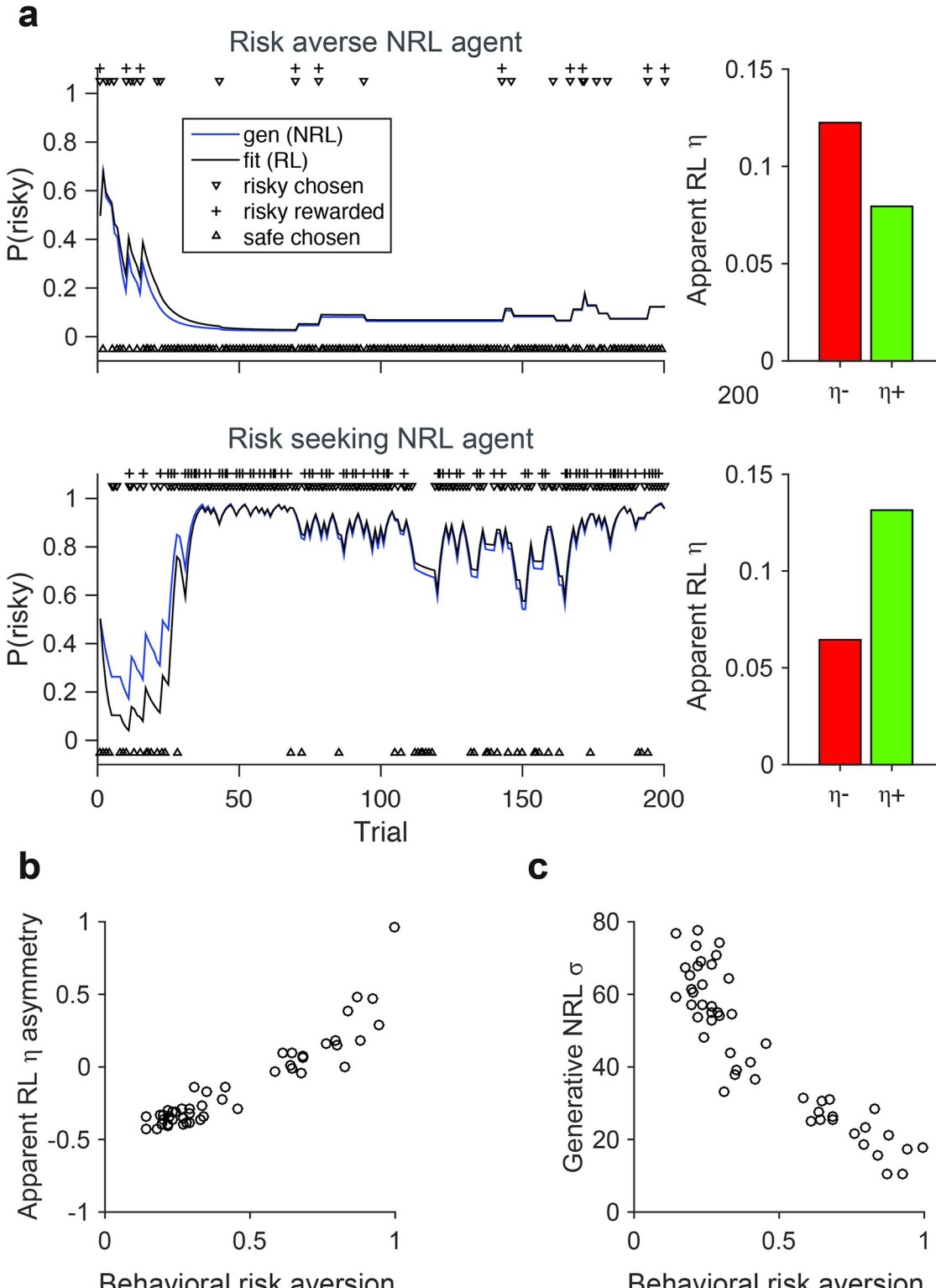

**Fig 3. NRL RPE asymmetry governs the degree of risk preference in reward learning. (a)** Examples of risk averse and risk seeking NRL agent behavior. Left, behavior in a task involving choices between a certain (100% chance of 20 A.U.) and a risky (50% chance of 0 or 40 A.U.) option. Blue lines, behavior of the generative NRL agent. Black lines, behavior of best fitting linear RL model with asymmetric learning rates. Right, apparent learning rates for negative and positive RPEs in linear RL model. Example risk averse $\sigma =$ 20 (top); example risk seeking $\sigma = 60$ (bottom). **(b)** Apparent relationship between risk preference and asymmetric learning rates

under assumption of linear reward coding. Behavioral risk aversion (percent choice of certain option) and learning rate asymmetry $(\eta^- \eta^+)/(\eta^- + \eta^+)$ defined as in previous work [26]. **(c)** Risk preference depends on RPE asymmetry in generative NRL model. Degree of behavioral risk aversion controlled by NRL semisaturation parameter.

Rather than the specific relationship between NRL parameterization and risk behavior, we emphasize that these results highlight the general finding that variability in risk aversion and implied learning rates can be generated by changes in the nonlinear NRL value function.

## A computational mechanism for distributional RL

Asymmetric NRL prediction errors also provide a computational mechanism for recent theories of distributional reinforcement learning. In contrast to standard RL approaches, where models learn a single scalar quantity representing the mean reward, distributional RL models learn about the full distribution of possible rewards [33,34]. The critical difference in distributional approaches is a diversity of RPE channels, with differing degrees of optimism about outcomes, which learn varying predictions about future reward. While most theoretical work relies on learning rate differences to produce RPE asymmetries, a recent report shows that dopamine neurons themselves exhibit sufficient characteristics to support distributional RL [35]: (1) dopamine neurons show a diversity of reversal points (reward magnitude where prediction errors switch from negative to positive), (2) dopamine neurons differ in their relative weighting of positive and negative RPEs, (3) this asymmetry in RPE weighting correlates with reversal point across neurons, and (4) the diversity in reversal points and asymmetries support a decoding of reward distributions. Notably, these findings show that distributional learning arises from asymmetries in RPE coding rather than in downstream learning rates, but do not address how such asymmetry might arise.

Here, we show that—given parametric diversity in individual dopamine neurons—NRL provides a computational mechanism for the neurophysiological characteristics supporting distributional RL. In contrast to standard RL, distributional RL posits that different RPE channels learn different value predictions in the identical reward environment. In a variable outcome environment [15,16], this predicts that individual dopamine neurons will exhibit different reversal points. Recent work shows that empirical dopamine responses exhibit this hypothesized variability in reversal points, driven by asymmetric scaling of negative and positive RPE responses [35]. Given that such asymmetric RPE responses are intrinsic to NRL, we examined the relationship between different NRL agents with varying $\sigma$ parameters and their reversal points in a stochastic outcome environment replicating the previously reported experiment [15,16]. Specifically, we simulated NRL responses to single rewards drawn from a distribution of seven different reward magnitudes (0.1, 0.3, 1.2, 2.5, 5, 10, or 20 A.U.); each possible reward outcome was drawn with an equal probability, and we examined steady state NRL RPE responses across outcomes. We find that NRL agents simulated in such a stochastic reward environment exhibit a diversity of reversal points (examples, Fig 4A) directly related to the individual NRL channel $\sigma$ parameter (Fig 4B).

Moreover, the intrinsic RPE asymmetry of NRL model units replicates key aspects of recorded dopamine neuron activity. First, NRL model units display a diversity of RPE biases, ranging from pessimistic (stronger negative RPE responses) to optimistic (stronger positive RPE responses) (Fig 4C). Second, the degree of RPE asymmetry is directly related to average expected value learned in the simulated environment (as quantified by reversal points; Fig 4D); this relationship mirrors that reported for recorded dopamine neurons. Finally, as a test of the ability of NRL to support distributional decoding, we examined whether a population of diverse NRL agents carries the necessary information to decode the distribution of previously experienced rewards (Fig 4E). Specifically, as done in recent work on empirical dopaminergic

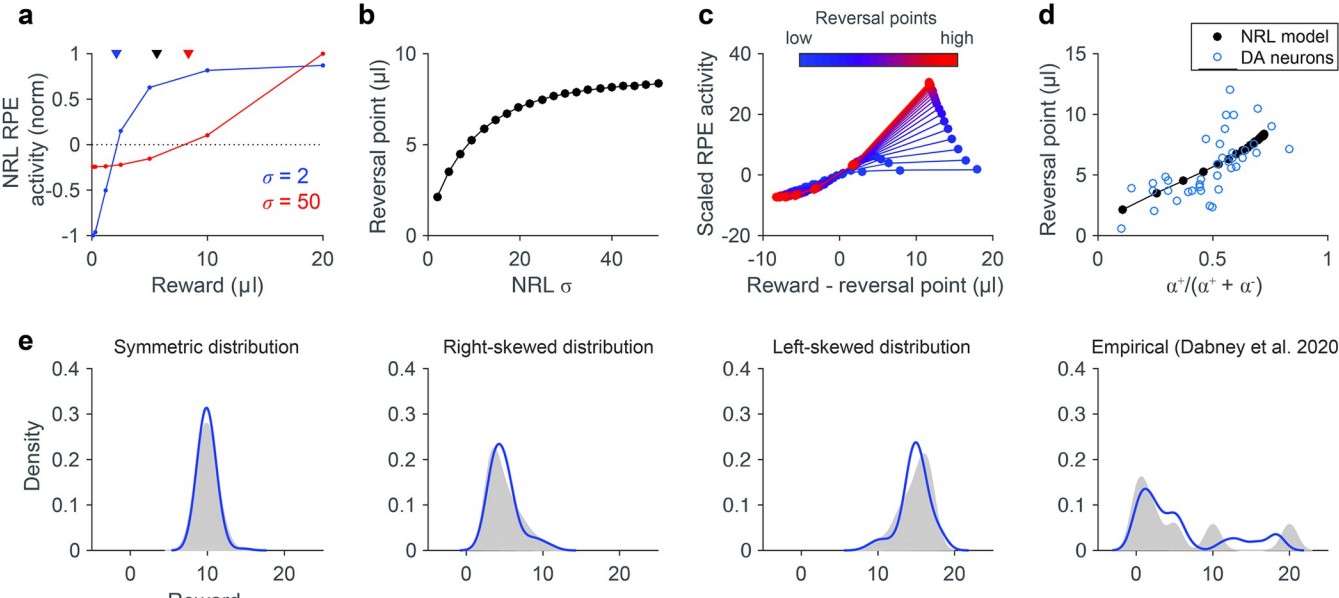

**Fig 4. NRL RPE asymmetry provides a computational basis for distributional reinforcement learning. (a)** Variable NRL RPE response asymmetries in a probabilistic reward environment. Examples show NRL agents with stronger negative (blue) and positive (red) RPE asymmetry. Note that these two agents exhibit different reversal points in the same reward environment (rewards = {0.1, 0.3, 1.2, 2.5, 5, 10, 20 μl}, as in previous work [35]). Triangles denote the true average reward (black) and estimated average reward learned by pessimistic (blue) and optimistic (red) NRL agents. **(b)** Learned reversal points vary systematically with NRL parameterization. RPE responses and reversal points quantified for varying $\sigma$ parameters. **(c)** Reversal points depend on NRL RPE asymmetry. Plots show NRL responses normalized by negative RPE slope and aligned to individual reversal points. As in empirical dopamine data, low (high) reversal points arise from stronger negative (positive) RPE asymmetry. **(d)** NRL asymmetry and learning match empirical dopamine data. Blue, dopamine neurons recorded in stochastic reward environment [35]; black, heterogeneous NRL agents in identical reward environment. Asymmetry is defined as in previous work as a function of positive ($\alpha^+$) and negative ($\alpha^-$) RPE coding slopes. **(e)** A population of NRL agents learns the distribution of experienced rewards. 40 NRL agents were simulated in four different reward environments: symmetric, right-skewed, left-skewed, and multimodal. Each panel plots the ground truth (gray) and decoded (blue) probability densities, with samples smoothed by kernel density estimation. Distribution decoding was performed via an imputation strategy, treating the NRL reversal points and response asymmetries as expectiles.

responses, we assumed that NRL reversal points and response asymmetries—learned in response to different reward environments—define a set of expectiles, and we transformed these expectiles into a probability density [35]. Following reward learning in a small population of NRL agents ($n = 40$, matched to empirical data), the probabilistic density of experienced rewards in different environments—including symmetric, asymmetric, or multimodal reward distributions–can be decoded from NRL responses (see Materials and Methods and S1 Appendix). Thus, NRL agents replicate the key empirical features seen in dopamine neurons: a diversity of reversal points, variable RPE asymmetries, a strong reversal point-asymmetry relationship, and an encoding of the statistical distribution of experienced rewards. Together, these findings suggest that the NRL algorithm provides a robust computational mechanism for distributional RL.

## Alternative parameterizations and biological plausibility

While for simplicity we parameterize NRL heterogeneity above via the semisaturation term $\sigma$, equivalent variability in RPE asymmetry can be generated by differential weighting of reward inputs:

$$V_{t+1} = V_t + \eta\left(\frac{(wR_t)^n}{\sigma^n + (wR_t)^n} - V_t\right) \tag{5}$$

Because an increase in input weighting is equivalent to a decrease in effective $\sigma$ (S1 Appendix), the $w$ term also controls RPE asymmetry: positive (negative) RPE biases occur for the same reward $R$ given sufficiently small (large) $w$ weights. In this alternative formulation, neural population diversity in RPE asymmetry arises simply from a variable weighting of reward inputs. Such variable input weighting is consistent with evidence for heterogeneity in both synaptic physiology and inputs to dopaminergic neurons [36]. Furthermore, differences in input weighting offers a more biologically plausible source of heterogeneity than the semisaturation term, which is typically considered a shared network property (i.e. baseline activity in a normalization pool) in circuit models of normalization [17,18]. For example, diversity in RPE asymmetry can be simply implemented by heterogeneity in the strength or number of reward-coding inputs to midbrain dopaminergic neurons.

Importantly, this alternative formulation also preserves the ability of the NRL model to capture adaptation, analogous to a reference point in history-dependent models of context-dependent decision-making. In sensory neuroscience, adaptation in neural responses is typically implemented in normalization approaches via a history-dependent $\sigma$ term that accounts for recent past stimuli. For example, normalization models with a dynamic $\sigma$ term that averages past contrast levels explains adapting responses in visual cortical neurons [17] and has been shown to improve efficient coding via redundancy reduction [37]. Furthermore, the adaptation of human valuation behavior is captured by a normalization mechanism with an equivalent $\sigma$ averaging past rewards [38]. Thus, by separating how the NRL algorithm models RPE asymmetry and reward history, this alternative parameterization can be used to examine neural and behavioral adaptation effects during reward learning.

## Discussion

Standard RL algorithms assume linear reward representations, but the brain represents objective rewards in a subjectively nonlinear manner; here we show that a nonlinear RL algorithm—employing the canonical divisive normalization computation—captures diverse neural and behavioral features of reward learning and decision-making. While the use of nonlinear reward transformations in RL is not novel, normalization generates both convex and concave regimes and, as a result, asymmetries in RPE responses for negative and positive prediction errors. In addition to matching empirical observations of saturating, nonlinear reward representations, the NRL model explain aspects of observed behavioral and neural data including variable risk preferences and asymmetric RPE responses thought to support distributional RL.

In its fullest form (Eq 5), the NRL algorithm is governed by three parameters with specific neurobiological interpretations and different functional implications. The exponent $n$ governs input amplification, and is generally viewed as a fixed property of normalization computations. The input weighting term $w$ implements parametric diversity in RPE asymmetries, which underlies the behavioral and neural diversity generated by the NRL model. Though a novel feature of the model, input weighting diversity may be simply realized as heterogeneity in synaptic strength or number of afferents. Finally, as in standard models of divisive normalization, a dynamic semisaturation term $\sigma$ can implement reference-dependent valuation by adjusting the normalized value response to recent rewards. The precise mechanism by which $\sigma$ adapts is not known, but possibilities include changes in pooled inputs contributing to normalization [17], long timescale nonlinear dynamics in divisive gain control [39, 40], and RL-like computations learning average reward. Regardless of mechanism, an adaptive $\sigma$ allows the NRL model to flexibly encode value in changing reward environments, adjusting reward coding relative to a reference point implemented as the $\sigma$ parameter.

Our results show that intrinsic asymmetries in reward learning can arise from subtleties in RPE coding rather than in downstream learning rates. However, these results do not preclude coexisting asymmetries in both systems. Both variability in dopaminergic genes and pharmacologic dopamine modulation affect reward learning in a valence-dependent manner, consistent with differential responses in striatal D1 and D2 receptors to positive versus negative RPEs [24,25]. Such downstream differences would drive differential weighting of prediction errors carried by dopaminergic inputs. On the other hand, increasing evidence shows valence-dependent differences in dopamine neuron responses [35,41], arguing for asymmetries in RPE coding itself. We suggest that NRL provides a computational mechanism to explain such asymmetries at the level of dopamine RPE representation, but valence-dependent biases in both prediction error coding and downstream processing likely play a role in biological learning.

While distributional RL has been proposed and implemented in computational algorithms, only recently has evidence arisen that dopamine neurons exhibit the necessary reward asymmetries [35]. Specifically, individual dopamine neurons differentially weight positive versus negative RPEs, and this relative valence-dependent weighting varies across neurons. However, how valence-dependent RPE coding arises is unknown. We show here that an RL system with a biologically-inspired normalized value function reproduces heterogeneous RPE asymmetries. In the NRL algorithm, asymmetry arises from the intrinsic curvature changes in the normalized reward function, and structured diversity in this asymmetry arises from parametric differences in normalized reward coding. Importantly, both asymmetry and structured diversity have plausible biological sources: normalization is produced by a number of mechanisms including feedforward inhibition, feedback inhibition, and synaptic depression [18], and asymmetry diversity requires only heterogeneity in input weighting (e.g. synaptic strength). Thus, RPE asymmetry in NRL arises solely from the reward processing circuit, and does not require differential weighting of separate sources of negative and positive RPE information.

Variability in RPE asymmetries is crucial to theories of distributional learning based on expectile regression, and empirical dopamine RPE asymmetries carry sufficient information to decode distributional information about experienced rewards via expectile-based decoding methods. Our results show that a population of NRL agents can learn and encode sufficient information to allow decoding of experienced reward distributions, at a level comparable to decoding from empirical responses [35]. However, while this shows that distributional information exists in both empirical and NRL responses, it is unclear whether expectile-based decoding is biologically plausible or employed by the brain. Alternative distributional codes have been suggested to be more computationally straightforward [42]; interestingly, these codes rely on variable nonlinear reward functions closely related to NRL responses, suggesting that NRL function in distributional learning may extend beyond expectile-based approaches. Further work is needed to verify distributional reward coding in downstream brain areas, identify the biological decoding algorithm, and test the contribution of NRL asymmetries to distributional learning.

Beyond capturing variability in risk preference and RPE asymmetry, the NRL model makes a number of further predictions about reward-guided behavior and neural activity. At the behavioral level, in addition to the across-subject variability shown here, the sigmoidal shape of the NRL value function predicts within-subject changes in risk preference. Under normalized value coding, the local curvature of the reward function depends on the relationship between rewards and the semisaturation term $\sigma$. Specifically, NRL predicts that individual risk preference should be magnitude dependent, with increasing risk aversion at larger outcomes; such outcome-dependent changes are consistent with some behavioral evidence [43,44], but remains to be tested in strict reinforcement learning scenarios. At the neural level, NRL

predicts an adaptive flexibility in individual (and population) dopamine neuron asymmetries. Unlike in other theories [35], RPE bias in a given NRL agent depends on the reward magnitude (relative to $\sigma$). Thus, while a population of NRL agents should retain their relative ranking of asymmetries in different environments, absolute asymmetries will change depending on the experienced rewards–an effect that may confer an advantageous adaptability. More broadly, when RPE asymmetry diversity is parameterized by input weights (Eq 4), the NRL algorithm can incorporate past reward information via a history-dependent $\sigma$ term. This suggests that NRL responses should capture contextual phenomena such as adaptive coding of reward values [45] and adaptation in risk preferences [46].

Finally, context-dependent valuation has been a recent area of interest in psychology, economics, and neuroscience, and it is important to consider how the NRL model relates to existing models of context-dependent valuation and choice. In its most detailed implementation (Eq 5), NRL provides a parametric control over both contextual adaptation (via $\sigma$) and RPE asymmetry (via $w$), implementing a reference-dependent S-shaped value function similar to several previously proposed models. Most broadly, prospect theory and related models propose an asymmetric value function convex in the region of losses and concave in the region of gains, with gains and losses determined relative to a reference point [47,48]. More recently, similar value functions arise in models derived from efficient coding principles in response to processing constraints [49] or designed to capture adaptation effects in choice behavior [50]. While the NRL value function also exhibits an S-shaped reference-dependence, it extends previous work in a number of ways. First, it applies nonlinear valuation to reinforcement learning, where linear reward functions are widely and conventionally employed; in contrast to prior behavioral theories, the NRL framework makes direct and testable predictions about both behavior and dopaminergic neural activity. Second, the normalized value function has a clear biological grounding: it defined from neurophysiological reward responses [15, 16, 21, 51], based on a canonical neural computation [18], and implementable in simple neural circuits [39, 40, 52]. Third, the NRL algorithm incorporates a novel parametric control of the degree and direction of asymmetric responding to negative and positive RPEs; diversity in NRL RPE asymmetry provides a unitary explanation for diversity in both behavior and neural responses. Finally, the combination of adaptation and distributional coding suggests the possibility of rich contextual reward learning, offering a single biological mechanism for diverse phenomena currently explained by different reference point centering [53], range adaptation [54], and range frequency based [55] models.

In summary, we present a model of reinforcement learning that incorporates a biologically-relevant nonlinear reward function implemented by divisive normalization. Normalized value coding introduces a parametrically tunable valence-based bias in prediction errors, and structured diversity in this bias captures both variable risk preferences across individuals and variable prediction error weighting across neurons. Together, these findings reconcile empirical and theoretical aspects of reinforcement learning, support the robustness of normalization-based value coding, and argue for the incorporation of biologically valid value representations into computational models of reward learning and choice behavior.

## Materials and methods

### NRL model

The NRL model applies a divisive normalization transform to experienced rewards (Eq 2), parameterized by an exponent $n$ and semisaturation term $\sigma$. Analytic analysis of NRL curvature is provided in the Supplementary Information. For all simulations (other than Fig 1c), we fixed $n = 2$. For simplicity, we implemented single state RL models that update value estimates

with the product of the RPE and learning rate $\eta$ (Eq 3); however, the NRL framework can be applied in more complicated models that incorporate features like action policy updating and temporal difference learning [1]. To examine RPE asymmetry, we quantified the response of different NRL agents (parameterized by varying $\sigma$; $\eta = 0.1$) to a fixed reward signal ($R = 50$ A. U.) corrupted with uniform noise (-40 to 40 A.U.). The degree of RPE asymmetry was quantified by piecewise linear regression analyses of negative and positive reward errors. Changing the parameters of the reward noise did not affect the qualitative finding of variable RPE asymmetries (S1 Appendix). All simulations and analyses were performed in MATLAB (R2015b).

### Risk-dependent decision-making

To examine risk preferences in the NRL model, we simulated agent behavior in a learning and choice task previously used to examine human subject behavior [26]. NRL agents were generated with random $\sigma$ parameters in the range [10,80]. In this task, each NRL agent chose between a certain option (100% chance of 20 A.U. reward) and a risky option (50% chance of 0 or 40 A.U. reward); initial values for both options were set to 0. In a given trial, choice was determined via a softmax function of estimated option values; to achieve a comparable level of choice stochasticity across different agents, the softmax temperature was inversely scaled with the $\sigma$ parameter. When chosen, the value of an option was updated based on received outcome according to Eq 3 ($\eta = 0.1$). For each NRL agent, we simulated behavior for 1000 trials. To examine how NRL agent behavior appears if linear reward functions are assumed, we fit NRL-generated behavior with an RL model with valence-dependent learning rates ($\eta^+$ and $\eta^-$; Eq 4) and quantified the apparent learning rate asymmetry $(\eta^- - \eta^+)/(\eta^- + \eta^+)$.

### Distributional RL

To examine whether information about experienced reward distributions were encoded in learned NRL responses, we quantified NRL agent behavior in different reward environments. Rewards were drawn from symmetric, left-skewed, and right-skewed distributions; in addition, we examined an environment with seven equiprobable rewards (0.1, 0.3, 1.2, 2.5, 5, 10, or 20 A.U.), replicating the conditions under which empirical dopamine neurons exhibit RPE asymmetries [15,16,35]. To facilitate comparison to empirical decoding performance, we examined 40 NRL agents with a diversity of semisaturation parameters (0.5 to 48 A.U.). For each environment, analytical steady state NRL functions and reversal points were calculated for each NRL agent (S1 Appendix), but similar results were obtained when NRL agents learned via sampling. Following identification of the reversal point, we estimated the RPE response asymmetry of each agent via separate linear regressions for negative and positive RPE responses around the reversal point.

Given a reversal point $R_n^*$ and RPE asymmetry $\tau_n$ for each NRL agent denoted by index $n$ (see S1 Appendix), distribution decoding was performed as previously described for empirical dopamine data [35]. Briefly, these data were interpreted as expectiles, where the $\tau_n$-th expectile had the value $R_n^*$. Decoding consisted of an imputation method to find a probability density that best matched the set of expectiles. As previously described, the density was parametrized as a set of 100 reward samples and optimization was performed to minimize a loss function between reversal points, asymmetries, and reward sample locations; see S1 Appendix for details.

### Code availability

MATLAB code used for simulation and analysis of the NRL model are available at https://osf.io/e6t5z/.

## Supporting information

**S1 Appendix. Analytical derivations and decoding methods.**
(DOCX)

## Acknowledgments

We thank W. Asaad, S. Bucher, P. Glimcher, A. Lee, M. Mehta, and B. Shen for comments on the manuscript.

## Author Contributions

**Conceptualization:** Kenway Louie.

**Investigation:** Kenway Louie.

**Writing – original draft:** Kenway Louie.

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
