## [Decision Letter · Decision Letter 0]

26 Mar 2022

Dear Dr. Louie,

Thank you very much for submitting your manuscript "Asymmetric and adaptive reward coding via normalized reinforcement learning" for consideration at PLOS Computational Biology.

As with all papers reviewed by the journal, your manuscript was reviewed by members of the editorial board and by several independent reviewers. In light of the reviews (below this email), we would like to invite the resubmission of a significantly-revised version that takes into account the reviewers' comments.

I enjoyed reading the paper, and the reviewer comments are pretty straightforward. One of the main questions is comparison to other models. You don't need to go on a wild goose chase exploring many different non-linear models, but it's important to convey to the reader more clearly what the space of possibilities is and why this particular model might be a good choice over other possibilities.

We cannot make any decision about publication until we have seen the revised manuscript and your response to the reviewers' comments. Your revised manuscript is also likely to be sent to reviewers for further evaluation.

Sincerely,

Samuel J. Gershman

Deputy Editor

PLOS Computational Biology

Reviewer's Responses to Questions

**Comments to the Authors:**

Reviewer #1: Thanks for the opportunity to review the paper titled “Asymmetric and adaptive reward coding via normalized reinforcement learning”. The paper proposes a divisive normalization model of subjective value to explain important processes characterising reinforcement learning. Simulations aim at demonstrating that the non-linear value function proposed by the model can explain (i) asymmetries between positive and negative learning rates and (ii) data about dopaminergic neurons. Overall, the paper is clearly written, and its contribution is timely and valuable. Below, please see some suggestions on how to improve the paper further:

Major points:

The author should clarify the specific contribution of the paper to the literature. It seems to me that the idea of proposing a reference-dependent sigmoid function (albeit in a different form) to describe subjective value is not novel (e.g., Woodford, 2012; Rigoli, 2019). The author should discuss this literature and clarify what the present paper adds to it. In my opinion, while previous literature adopting similar sigmoid models focuses on reference effects, the novel contribution of the paper is to explore the implications of a sigmoid value function in the context of RL; and show how this can explain (i) reported asymmetries in learning rate and (ii) recent data about dopamine function. Does the author agree on this interpretation? In any case, it would be helpful to clarify the precise contribution of the paper.

Although the paper hints to reference effects in shaping the value function (also consistent with the literature on normalization mentioned in the introduction), the nature of these effects is not explained. Specifically, the paper is silent about where the parameters sigma and n come from. Do they depend on the property of the distribution of rewards (e.g., sigma being an average and n being an index of variability)? I think that an in-depth analysis of this aspect is not necessary here, as the main points of the paper emerge clearly enough as it is. Yet, a short clarification of this important aspect of the model would be beneficial. Relatedly, given that the focus is on RL, the discussion might briefly speculate on how the parameters are themselves acquired via learning.

As argued above, the simulation described in fig. 3 appears crucial to me, as it reflects one of the main contributions of the paper (explaining asymmetries in learning rates as arising from non-linear value functions). However, I am not sure how robust the result that sigma inversely correlates with risk aversion is. Does this also depend on the range of rewards simulated? I suggest to vary the reward distribution (keeping the parameters fixed) in different simulations and show what happens. In general, a few other simulations on this point (e.g., based on empirical studies conducted by Palminteri’s lab or Frank’s lab) would make the argument more convincing. Relatedly, what is the role of the parameter n in risk sensitivity?

Regarding the paragraph starting with “Recent work shows that empirical dopamine responses exhibit…”, I found the description of the simulation unclear. Please provide all the relevant information about this simulation in this paragraph.

Minor points:

Fig 1 b: please report the values of sigma and n adopted in the simulation.

Line 100: I think it should be “asymmetric prediction errors”

“Intuitively, these apparent learning rates arise because fitting with the standard RL model (Eqn. 4) approximates bipartite linear regression on the nonlinear value function”. This sentence is not clear to me.

Regarding the paragraph “Moreover, the intrinsic RPE asymmetry of NRL model..” the author nicely draws a parallel between the simulation’s results and the main four findings reported by Dabney at al (2020) outlined previously. It would help to make the parallel more explicit.

References

Rigoli, F. (2019). Reference effects on decision-making elicited by previous rewards. Cognition, 192, 104034.

Woodford, M. (2012). Prospect theory as efficient perceptual distortion. American Economic Review, 102(3), 41-46.

Reviewer #2: Review of Asymmetric and adaptive reward coding via normalized reinforcement learning by Louie.

Summary

In the present study, the author describes behavioral and computational implications of using non-linear reinforcement learning (RL) models. The author characterized a RL model implementing a non-linear value function of reward via a divisive normalization computation. The author shows that this non-linear specification can produce asymmetrical prediction errors coding and then explain both behavioral phenomena such as attitudes toward risk and computational mechanisms underlying distributional learning in the brain. The paper is well written, clear, and concise, and the question raised by the author is of prime interest. The link made between non-linear value functions and updating asymmetry in linear models with multiple learning rates is particularly interesting and promising, and in making that link and describing its computational implication, the study fulfils its goal perfectly. The paper is very informative, and I enjoyed reading it. On a side note, it is not clear how the model, in its current version, will be useful to analyze behavior in future studies. The later can produce relevant effects observed in humans but the lack of clear view on what could represent (cognitively) or determine the semisaturation parameter is problematic. It is not obvious either whether the model can fit reliably human behavior and make actual predictions about the later.

Major comments

#1 Alternative non-linear models

The non-linear valuation in the model can produce asymmetric prediction errors coding, but so would do other nonlinear, convex or concave, utility functions. For instance, another function that is convex then concave (like the present function for n≥2) is the prospect theory valuation function, which is convex below the reference point and concave above. Moving the reference point would then produce similar effects as moving the semisaturation parameter in the present model. Beyond its explanatory power at the neural level, could author explain the advantages of using the present model over other non-linear ones?

I understand why the divisive normalization is used here but is it, in the present version, a good behavioral model? Regarding the prospect theory function, moving arbitrarily the reference point to account for various behaviors, without clear assumption about how it is set, would not make much sense. And the same could be said with moving the semisaturation parameter in the present model, which has furthermore a less clear definition. Could the author comment on this point?

#2 Interpretation of the semisaturation parameter

Related to point #1, I find it hard to grasp the meaning of the semisaturation parameter, at the behavioral level. In the simulations presented in the paper, it seems that its only goal is to place the concave or convex portion of the valuation curve adequately on the range of objective values in order to generate a positive or a negative asymmetry in prediction error coding respectively. In the same way and as an example, does it make sense to define the semisaturation parameter as being twice as large as the upper bond of the range of objective value in the considered environment (e.g., the choice task described here)?

3# On the use of the model in behavioral studies

It is not clear from the paper how well the model could fit behavioral data and how well the parameters could be recovered from simulated data. For instance, in the behavioral part of the paper, the model is used to generate data in a choice task, could the simulated data be reasonably well fitted by the generative model and the parameters of the latter well recovered?

Minor comments

#1 Figure 1b - parameters

It would be useful for readers to make explicit parameters that have been used in figure 1b.

#2 Figure 3a parameters

Similarly, it would be useful for readers to know which parameters have been used to generate the risk-seeking and risk-averse agents from figure 3a. We know the range of values that has been used for all agents but not the values of these two particular cases.

**Have the authors made all data and (if applicable) computational code underlying the findings in their manuscript fully available?**

Reviewer #1: Yes

Reviewer #2: Yes

PLOS authors have the option to publish the peer review history of their article (what does this mean?). If published, this will include your full peer review and any attached files.

Reviewer #1: **Yes: **Francesco Rigoli

Reviewer #2: No
---

## [Decision Letter · Decision Letter 1]

1 Jul 2022

Dear Dr. Louie,

We are pleased to inform you that your manuscript 'Asymmetric and adaptive reward coding via normalized reinforcement learning' has been provisionally accepted for publication in PLOS Computational Biology.

Before your manuscript can be formally accepted you will need to complete some formatting changes, which you will receive in a follow up email. A member of our team will be in touch with a set of requests. Also please note a couple of minor suggestions from Reviewer 1.

Best regards,

Samuel Gershman

Deputy Editor

PLOS Computational Biology

Reviewer's Responses to Questions

**Comments to the Authors:**

Reviewer #1: Thanks again for the opportunity to review the paper. The authors have addressed all the points I highlighted, and the paper appears as a valuable contribution to the literature. I have two further minor suggestions:

- the sentence "the adaptation of human valuation behavior is captured by a normalization

mechanism with an equivalent σ averaging past rewards (38)" sounds a bit overstated; I suggest to add something like "at least in some circumstances"

- in the response to reviewers, the authors mention the empirical literature about asymmetrical RL, and argue that an analysis of this is beyond the scope of the manuscript - this is something left for future work. The authors' reply sounds reasonable. However, I think it is preferable to briefly acknowledge this in the discussion (and potentially briefly speculate about how the model could capture some of the main findings)

Reviewer #2: I thank the author for their pertinent and precise answers to my comments. The author performed new analyses and added new paragraphs to the manuscript, improving further an already very interesting study.

**Have the authors made all data and (if applicable) computational code underlying the findings in their manuscript fully available?**

Reviewer #1: None

Reviewer #2: Yes

PLOS authors have the option to publish the peer review history of their article (what does this mean?). If published, this will include your full peer review and any attached files.

Reviewer #1: **Yes: **Francesco Rigoli

Reviewer #2: No

---

## [Editor Report · Acceptance letter]

11 Jul 2022

PCOMPBIOL-D-22-00336R1 

Asymmetric and adaptive reward coding via normalized reinforcement learning

Dear Dr Louie,

I am pleased to inform you that your manuscript has been formally accepted for publication in PLOS Computational Biology. Your manuscript is now with our production department and you will be notified of the publication date in due course.

With kind regards,

Zsofia Freund
